# Carbon Dots for Killing Microorganisms: An Update since 2019

**DOI:** 10.3390/ph15101236

**Published:** 2022-10-08

**Authors:** Fengming Lin, Zihao Wang, Fu-Gen Wu

**Affiliations:** State Key Laboratory of Bioelectronics, School of Biological Science and Medical Engineering, Southeast University, 2 Sipailou Road, Nanjing 210096, China

**Keywords:** antibacterial, bactericidal, disinfection, carbon nanodots, carbonized polymer dots

## Abstract

Frequent bacterial/fungal infections and occurrence of antibiotic resistance pose increasing threats to the public and thus require the development of new antibacterial/antifungal agents and strategies. Carbon dots (CDs) have been well demonstrated to be promising and potent antimicrobial nanomaterials and serve as potential alternatives to conventional antibiotics. In recent years, great efforts have been made by many researchers to develop new carbon dot-based antimicrobial agents to combat microbial infections. Here, as an update to our previous relevant review (C 2019, 5, 33), we summarize the recent achievements in the utilization of CDs for microbial inactivation. We review four kinds of antimicrobial CDs including nitrogen-doped CDs, metal-containing CDs, antibiotic-conjugated CDs, and photoresponsive CDs in terms of their starting materials, synthetic route, surface functionalization, antimicrobial ability, and the related antimicrobial mechanism if available. In addition, we summarize the emerging applications of CD-related antimicrobial materials in medical and industry fields. Finally, we discuss the existing challenges of antimicrobial CDs and the future research directions that are worth exploring. We believe that this review provides a comprehensive overview of the recent advances in antimicrobial CDs and may inspire the development of new CDs with desirable antimicrobial activities.

## 1. Introduction

Owing to the long-term use and overuse of antibiotics, pathogens have become resistant to almost all existing traditional antibiotics by mutating or acquiring drug-resistant genes from other organisms. It is urgently necessary to develop novel effective antimicrobial compounds as potent alternatives to the conventional small-molecule antibiotics to address the issue of microbial drug resistance. The great advancement of nanoscience and nanotechnology has offered a new solution for the development of antimicrobial materials. Several types of nanomaterials are known to exhibit antibacterial properties. Particularly, inorganic metal and metal oxide nanoparticles have been intensively investigated for their potential use as antimicrobial agents [1,2]. Although these metal (e.g., Au and Ag) and metal oxide (e.g., Fe_2_O_3_, CuO, and ZnO) nanoparticles possess antimicrobial activities, the release of metal ions may cause nonspecific biological toxicity, which urgently requires the development of safer antimicrobial nanomaterials [3,4]. Among the large variety of antimicrobial nanomaterials, carbon dots (CDs) have received ever-increasing attention, mainly due to their easy preparation and functionalization, great water dispersity, and satisfactory biocompatibility. One appealing merit for CDs is that their property and function can be easily manipulated during the synthesis or post-modification stage, which is highly useful for antibacterial applications. CDs are zero-dimensional carbonaceous nanoparticles with sizes no more than 20 nm, also termed “carbonized polymer dots”, “carbon quantum dots”, or “carbon nanodots” [5,6,7,8,9,10,11]. CDs can be prepared from a wide variety of natural materials such as biomass and waste, and a huge array of chemical agents [12,13]. There are two well-known CD preparation strategies: bottom-up strategy and top-down strategy. The synthetic approaches for CDs include hydrothermal/solvothermal reaction, pyrolysis, sonication, microwave irradiation, etc. [12]. The broad applications of CDs in sensing [12,14,15,16], optoelectronics [17], energy [18], catalysis [12,19], and nanomedicine [20,21,22,23,24], have been demonstrated since their discovery in 2004 [25].

Currently, three antimicrobial mechanisms have been reported for CDs, including cell wall/membrane disruption, reactive oxygen species (ROS) generation, and DNA damage [23]. The inhibitory action of CDs on microorganisms depends on the composition, size, shape, and surface chemistry of CDs. It is extremely difficult to explain the antimicrobial mechanisms of CDs without performing careful structural characterizations of the CDs. Specifically, the catalytic activity, the crystallographic structure, the surface state (defect or functionalization), and charge transfer are important factors that contribute to the antimicrobial activity of CDs. However, currently, except for the several studies that mentioned the effect of surface functionalization on the antimicrobial activity of CDs [7,24,26], detailed evaluations of the other factors are still lacking in the current CD-based antimicrobial studies. As a result, more attention should be paid to the investigations of the effect of the other factors on the antimicrobial activity of CDs in the future.

In 2019, we have reviewed the advancements of CDs in sensing and killing microorganisms in terms of their preparation, functionalization, toxicity, and underlying antimicrobial mechanism [16]. Nevertheless, the past three years have witnessed the booming applications of CDs in the antimicrobial field. Hence, an update on this topic is essential to embrace the latest progress in this field. Numerous CDs have been reported for antimicrobial therapy in recent years, and they can be classified into four types including nitrogen-doped CDs, metal-containing CDs, antibiotic-conjugated CDs, and photoresponsive CDs (Figure 1). We discuss their raw materials, synthetic approaches, modification methods, antimicrobial abilities, and the related antimicrobial mechanisms if available in detail below. Furthermore, we also introduce the advances in the applications of the antimicrobial CDs in industry and medicine. Finally, we discuss the current limitations of antimicrobial CDs and propose some research directions. 

## 2. Antimicrobial CDs

### 2.1. Nitrogen-Doped CDs 

#### 2.1.1. Nitrogen-Doped CDs Derived from Biomass

When we prepared our previous review in 2019, a large number of fluorescent CDs synthesized from different natural sources had been harnessed for microbial imaging, but few CDs derived from biomass had been utilized as antimicrobial agents [16]. Nevertheless, in the last three years, the development of green synthetic methods for fabricating antimicrobial CDs from different natural carbon precursors has attracted considerable interest (Table 1), and such a method is facile, cost-effective, and eco-friendly. CDs prepared from *Lawsonia inermis* (Henna) [27], oyster mushroom (*Pleurotus species*) [28], osmanthus leaves [29], tea leaves [29], *Ananas comosus* waste peels [30], *Impatiens balsamina* L. stems [31], *Aloe vera* leaves [32], medicinal turmeric leaves (*Curcuma longa*) [33], rosemary leaves [34], sugarcane bagasse pulp [35], waste tea extract [36], waste jute caddies [37], *Forsythia* [38], and *Artemisia argyi* leaves [39], have been reported to possess antimicrobial activities. It is interesting to find that all these biomass-derived CDs contain the nitrogen element, which might be from the proteins, animo acids, and nucleic acids in biomass. For instance, Wang et al. synthesized CDs (ACDs) from *Artemisia argyi* leaves through a smoking simulation approach (Figure 1A) [39]. ACDs have spherical morphology with a diameter of 2–5 nm (Figure 1B). ACDs displayed selective antibacterial ability toward Gram-negative bacteria like *Escherichia coli* (*E. coli*), ampicillin-resistant *E. coli* (ARE *E. coli*), kanamycin-resistant *E. coli* (KRE *E. coli*), *Pseudomonas aeruginosa* (*P. aeruginosa*), and *Proteusbacillus vulgaris* (*P. vulgaris*), but not Gram-positive bacteria such as *Staphylococcus aureus* (*S. aureus*) and *Bacillus subtilis* (*B. subtilis*) (Figure 1C). ACDs could kill 100% Gram-negative bacteria at 150 μg mL^−1^. The antibacterial mechanism study demonstrated that ACDs could only disrupt the cell walls of *E. coli* rather than those of *S. aureus*, since according to the high magnification images in (d and h), only the *E. coli* cells showed shrunken and damaged cell structures (Figure 1D). In addition, ACDs could change the secondary structure and thus the activity of cell wall-related enzymes in Gram-negative bacteria. More interestingly, ACDs could strongly prevent the biofilm formation of *E. coli*. The development of ACDs is of great value for the treatment of infections associated closely with Gram-negative bacteria. Saravanan et al. prepared CDs by one-step hydrothermal treatment of medicinal turmeric leaves (*Curcuma longa*) [33]. The as-prepared CDs exhibited antibacterial activities toward both Gram-negative bacteria (*E. coli* and *Klebsiella pneumoniae* (*K. pneumoniae*)) and Gram-positive bacteria (*S. aureus* and *Staphylococcus epidermidis* (*S. epidermidis*)) due to their release of ROS. Collectively, these naturally derived CDs from biomass represent potent candidates as new antimicrobial agents to combat antibiotic-resistance of microorganisms. 

#### 2.1.2. Nitrogen-Doped CDs Derived from Nitrogen-Containing Compounds 

In addition to biomass, nitrogen-doped CDs can also be prepared from nitrogenous compounds such as proteins [40], amino acids [41,42,43,44], natural amines [45,46,47], quaternized compounds [48,49], polyethyleneimine (PEI) [50,51,52,53], diethylenetriamine (DETA) [45,54], 2,2′-(ethylenedioxy)-bis(ethylamine) [55], *p*-phenylenediamine [56], and *m*-phenylenediamine [57] for antimicrobial purposes. 

Nitrogen-doped CDs have been obtained using protein (protamine sulfate) as the raw material. Zhao et al. reported the simple and fast synthesis of multifunctional blue-emitting protamine sulfate (PS)-based CDs (PS-CDs) by a one-step microwave-mediated approach (Figure 2A) [40]. PS-CDs featured great antibacterial efficacy against *S. aureus* with a minimum inhibitory concentration (MIC) of 25 μg mL^−1^ and methicillin-resistant *S. aureus* (MRSA) with an MIC of 37.5 μg mL^−1^. Furthermore, PS-CDs displayed high water-dispersity, low cytotoxicity, and excellent blood compatibility. The authors also revealed the antibacterial mechanism of PS-CDs: PS-CDs bound to bacteria by electrostatic interaction, damaged cell membrane, entered the cells, and disrupted the normal survival functions of the bacteria. To conclude, this work employs protein as a raw material to prepare CDs that can be internalized easily by bacteria to realize bactericidal effect. 

Several CDs prepared from amino acids such as arginine [41], alanine [42,43], L-tryptophan [58,59], cysteine [43,44], lysine [59], and arginine [59] have been reported to possess antimicrobial activities. Suner et al. synthesized nitrogen-doped arginine CDs (termed Arg CDs) utilizing citric acid as the carbon source and arginine as the amine source by a microwave-mediated approach (Figure 2B) [41]. Arg CDs displayed an MIC of 6.250 mg mL^−1^ against *S. aureus*. To enhance the antimicrobial activity of Arg CDs, two nanocomposites, Arg-Ag CDs and Arg-Cu CDs, were synthesized by generating Ag and Cu nanoparticles (NPs) within Arg CDs. Arg-Ag CDs and Arg-Cu CDs exhibited an MIC of 0.062 and 0.625 mg mL^−1^ against *S. aureus*, respectively. In addition, Arg-Ag CD possessed 0.125 and 0.312 mg mL^−1^ minimum bactericidal concentration (MBC) values against *S. aureus* and *E. coli*, respectively. Pandey et al. prepared CDs from citric acid and β-alanine through a microwave-mediated method [42]. The as-synthesized CDs repressed the growth of diverse Gram-negative bacteria such as *E. coli*, *Salmonella*, *Pseudomonas*, *Agrobacterium*, and *Pectobacterium* species.

Besides proteins and amino acids, other natural amines have been explored as precursors to construct antimicrobial CDs, such as histamine [45], cadaverine [45], putrescine dihydrochloride [45], spermine tetrahydrochloride [45], L-glutathione [46], and spermidine [47]. As an example, Hao et al. prepared positively charged CQDs (PC-CQDs) from citric acid and L-glutathione (Figure 2C) [46]. PC-CQDs exhibited high antibacterial activity against *S. aureus*, *E. coli*, *P. aeruginosa,* and MRSA. PC-CQDs strongly attached to the bacterial cell surfaces due to their small size and the surface groups –NH_2_ and –NH, entered the cells, and induced the conformational change of DNA and the production of ROS, leading to the rupture of the bacterial cells. It is worth noting that PC-CQDs did not cause detectable drug resistance or hemolysis. No drug resistance was observed in *S. aureus*, *E. coli*, and MRSA incubated with PC-CQDs for over 30 days. Moreover, PC-CQDs were deployed for the antibacterial treatment of mixed *S. aureus-* and *E. coli*-infected wounds in rats with low in vivo toxicity, showing the same therapeutic effect as the traditional antibiotic levofloxacin hydrochloride.

In addition to natural amines as introduced above, other nitrogen-containing agents have also been chosen to prepare nitrogen-doped CDs as antimicrobial agents, including quaternized compounds [48,49], PEI [50,51,52,53], DETA [45,54], 2,2′-(ethylenedioxy)-bis(ethylamine) [55], *p*-phenylenediamine [56], *m*-phenylenediamine [57], etc. Our group prepared quaternized CDs through one-step solvothermal treatment of glycerol and dimethyloctadecyl[3-(trimethoxysilyl)propy]ammonium chloride (Si-QAC) for selective Gram-positive bacterial inactivation (Figure 2D) [48]. The synthesized quaternized CDs could selectively interact with the Gram-positive bacteria due to the distinct surfaces of Gram-positive and Gram-negative bacteria. The quaternized CDs possessed a zeta potential of +33 mV, ensuring their successful electrostatic interaction with negatively charged bacterial cells, while the presence of long alkyl chains in the CDs enabled them to interact with the bacterial cells via hydrophobic interaction. Therefore, the CDs could firmly adhere to (or insert into) the bacterial cell surface which changed the charge balance of the bacterial surface, resulting in the inactivation of Gram-positive bacteria both in vitro and in vivo. At the same time, the CDs featured strong fluorescence emission, which was utilized for fast Gram-type identification. In this way, the CDs may hold the potential for treating Gram-positive bacteria-caused infections. More interestingly, in a later study, we have demonstrated that the CDs showed excellent biofilm penetration capacity due to their small size and could effectively inhibit the biofilm formation and eradicate the formed biofilms [22]. Thus, the CDs represent a highly efficient strategy to combat biofilm-involved infections. In another report, Li et al. fabricated different types of polyamine-modified carbon quantum dots (CQDs) including CQD_600_, CQD_1w_, and CQD_2.5w_, via simple hydrothermal treatment of citric acid and branched PEI (bPEI) with different molecular weights [50]. CQD_2.5w_ possessed higher antibacterial and antibiofilm activities against *S. aureus* and *E. coli* than CQD_1w_ and CQD_600_, since the larger molecular weight of bPEI yielded a larger amount of protonated amines on the surface of the CQDs, giving rise to enhanced electrostatic interaction between CQDs and bacterial cells, and the longer surface corona of the CQDs making their penetration into biofilms easier. Additionally, all the three CQDs had negligible cytotoxicity. However, the reason why longer surface corona of CQDs can give rise to their easier penetration into biofilm was not explained, which requires future investigation. Further, Zhao et al. developed nitrogen-doped CDs (NCQDs) as an antimicrobial nanoagent against *Staphylococcus* to treat infected wounds [54]. NCQDs were made from D(+)-glucose monohydrate and DETA by a heat fusion method. NCQDs displayed antibacterial activity toward *Staphylococcus*, especially against MRSA. Transmission electron microscopy (TEM) analysis showed that NCQDs could damage the cell structures of *S. aureus* and MRSA, but not *E. coli*. NCQDs exhibited the same therapeutic effect as vancomycin in the treatment of MRSA-infected wounds with negligible toxicity to the main rat organs such as heart, kidney, liver, lung, and spleen.

### 2.2. Metal-Containing CDs

There are different types of metal-containing CDs that can be used as antimicrobial agents: metal ion-doped CDs [36,60,61,62], metal nanoparticle-decorated CDs [41,60,63,64,65], CD/metal oxide nanocomposites [66,67,68,69,70], and CD/metal sulfide nanocomposites [71]. First, CDs doped with metal ions such as Cu^2+^ [36] and Ag^+^ [60] have been explored as antimicrobial materials. Qing et al. reported water-dispersible Cu^2+^-doped CDs (Cu^2+^–CDs) for antibacterial application [36]. Cu^2+^–CDs were prepared by one-step hydrothermal carbonization of cupric acetate monohydrate (Cu(Ac)_2_•H_2_O) and waste tea extract. Cu^2+^–CDs showed an inhibitory effect on *S. aureus* with an MIC of 0.156 mg/mL. Moreover, Cu^2+^–CDs possessed low cytotoxicity and appealing biocompatibility. Second, CDs can serve as reducing and stabilizing agents to generate metal nanoparticles on their surface, forming metal nanoparticle-modified CDs for killing microorganisms [41,60,63,64,65]. For instance, antimicrobial silver nanoparticle-decorated CDs (CD-2) were prepared using a two-step method, in which CDs (PEI-CD) were first obtained by hydrothermal treatment of PEI, and Ag^+^ was then reduced to Ag NPs in the presence of formaldehyde and the resultant Ag NPs were bound onto the surface of PEI-CD to produce CD-2 (Figure 3A) [60]. By inducing membrane disruption and intracellular DNA/protein damage, CD-2 displayed high and broad-spectrum antimicrobial activities against Gram-positive bacteria (*S. aureus*), Gram-negative bacteria (*E. coli*, *P. aeruginosa*, and *P. vulgaris*), and fungi (*S. cerevisiae*), with excellent biocompatibility. Third, CDs were integrated with metal oxide to form CD/metal oxide nanocomposites [66,67,68,69,70]. As an example, Gao et al. fabricated CD/ZnO/ZnAl_2_O_4_ nanocomposites which possessed an excellent antibacterial property with antibacterial ratios of 97% and 94% against *S. aureus* and *E. coli*, respectively [68]. Within the antibacterial concentration range, the nanocomposites were nontoxic to human cells. As shown in Figure 3B, the authors proposed a dual-mode antibacterial mechanism for the nanocomposite. On the one hand, the binding of the CD/ZnO/ZnAl_2_O_4_ nanocomposites to the surface of the bacteria blocked the channels of the bacterial nutrient supply from the environment, accelerating apoptosis in the bacteria. On the other hand, the nanocomposites could generate singlet oxygen that damages DNA/RNA, proteins, and phospholipids in the bacterial cells. Also, the presence of CDs in the nanocomposites resulted in strengthened electrostatic interaction between the nanocomposites and the bacteria, and increased singlet oxygen production, which enhanced the bacterial elimination effect of the nanocomposite. Lastly, CD/metal sulfide nanocomposites have been constructed for combating microorganisms. Gao et al. synthesized carbon quantum dots (CQDs) through aldol polymerization reaction using acetone as the carbon source (Figure 3C) [71]. Then CQDs/Ag_2_S/CS nanocomposites were prepared through an in situ growth method using polyvinylpyrrolidone (PVP) as the crosslinking agent. The CQDs/Ag_2_S/CS nanocomposites exhibited excellent antibacterial property against *E. coli* and *S. aureus* with an MIC of 0.1 mg/mL, and against MRSA with an MIC of 0.25 mg/mL. The CQDs/Ag_2_S/CS nanocomposites strongly bound to the surface of the bacteria, leading to the destruction of cell wall and cell membrane and inducing the bacterial cell death (Figure 3C). Notably, no drug resistance was observed for the CQDs/Ag_2_S/CS nanocomposites.

### 2.3. CDs Derived from Antibacterial Compounds (Including Antibiotics)

Antimicrobial CDs have also been developed by using traditional antibiotics and common antibacterial compounds as the raw materials to trade the old for the new, such as kanamycin sulfate [72], levofloxacin hydrochloride [73,74], quaternary ammonium compounds [22,48,49,75,76], and 2,4-dihydroxybenzoic acid [77]. Luo et al. prepared CDs (termed CDs-Kan) from kanamycin sulfate by a one-step hydrothermal method [72]. CDs-Kan were demonstrated to preserve the main bactericidal functional groups of kanamycin like the amino sugar and amino cyclic alcohol, which ensured their good antibacterial activity. Specifically, CDs-Kan inhibited the growth of *E. coli* and *S. aureus* with good biocompatibility. Wu et al. reported cationic levofloxacin-derived CDs (LCDs) with enhanced antibacterial activities and low drug resistance (Figure 4A) [74]. LCDs were synthesized from levofloxacin hydrochloride through a simple one-pot hydrothermal method. With the preservation of the active groups from levofloxacin, LCDs featured notable bactericidal activity against *S. aureus* and *E. coli* with an MIC of 0.125 μg/mL, which was lower than that of levofloxacin hydrochloride. It is worth noting that LCDs displayed low drug resistance, good aqueous dispersity, and outstanding biosafety, while retaining the broad-spectrum antibacterial activity of levofloxacin. In addition to *S. aureus* and *E. coli*, LCDs could kill other microorganisms including MRSA, *Enterococcus faecalis* (*E. faecalis*), *S. epidermidis*, *Listera monocytogenes* (*L. monocytogenes*), *P. aeruginosa*, and *Serratia marcescens*. LCDs could enter the bacterial cells via electrostatic interaction between the positively-charged LCDs and the negatively-charged bacteria. Once entering the bacterial cells, LCDs produced ROS to destroy cell membrane and the normal state of bacteria, causing cell death. LCDs were deployed for the treatment of bacteria-infected wounds and pneumonia in mice with enhanced therapeutic efficacy without harming normal tissues as compared to levofloxacin. Zhao et al. fabricated quaternary ammonium carbon quantum dots (QCQD) from 2,3-epoxypropyltrimethylammonium chloride and diallyldimethylammonium chloride via the hydrothermal reaction (Figure 4B) [75]. QCQD displayed admirable bactericidal effect toward Gram-positive bacteria, including *S. aureus*, MRSA, *E. faecalis*, *L. monocytogenes*, and *S. epidermidis*. QCQD also featured satisfactory biocompatibility as demonstrated by in vivo and in vitro toxicity assays. Thus, QCQD were successfully utilized in the treatment of MRSA-infected pneumonia in mice, prompting the regression of pulmonary inflammation in the mouse lung. As revealed by the quantitative proteomics, the antibacterial ability of QCQD could be attributed to the fact that QCQD might mainly act on ribosomes and upregulate the proteins involved in RNA degradation, causing interference to the protein translation, posttranslational modification, and protein turnover in bacterial cells.

Meanwhile, CDs can be employed as the carriers of existing antibiotics to realize the controlled release of these antibiotics. Saravanan et al. prepared N@CDs by hydrothermal treatment of *m*-phenylenediamine (Figure 4C) [57]. N@CDs exhibited antibacterial activities against *E. coli* and *S. aureus* with an MIC of 1 and 0.75 mg/mL, respectively. Additionally, N@CDs were applied as nanovehicles for sustained time-dependent release of the traditional antibiotic ciprofloxacin in the physiological condition.

### 2.4. Photoresponsive CDs

#### 2.4.1. Photodynamic Therapy (PDT)

In PDT, photosensitive agents (photosensitizers) are sensitized by light in the presence of oxygen to generate ROS such as free radicals and singlet oxygen [78,79]. The produced ROS can break DNA, inactivate enzymes, and oxidize amino acids, resulting in cell necrosis/apoptosis. PDT represents a promising alternative to antibiotics in killing microorganisms, because of its fascinating advantages such as high spatial controllability, antibiotic resistance independence, and low cumulative toxicity. The photo-generated electrons and holes of CDs that are related with PDT action mechanism entail various catalytic processes [80]. Meanwhile, CDs have a relatively wide visible spectral region. These properties enable CDs to be promising antibacterial photosensitizers. After irradiation with light of a given wavelength, some bare CDs can produce ROS that are capable of inactivating microorganisms, and these CDs can thus serve as potent antimicrobial photosensitizers.

Beyond bare CDs fabricated from mushroom [81], graphite rods [82], and the poloxamer Pluronic F-68 [83], as summarized in our previous review [16], more bare CDs have been reported to possess intrinsic photodynamic characteristics, including graphitic carbon nitride quantum dots (g-CNQDs) [84], red-emitting CDs (R-CDs) [77], CQDs constructed from citric acid and 1,5-diaminonaphthalene by solvothermal reaction [85], nitrogen- and/or sulfur-doped CDs derived from amino acids [43], nitrogen and iodine co-doped CDs (N/I-CD) prepared by hydrothermal treatment of iohexol [86], metal-doped CDs such as zinc-doped CDs [87], copper-doped CDs [88], and terbium-doped CDs [89]. Yadav et al. constructed green-fluorescent g-CNQDs from melamine and ethylene diamine tetraacetic acid (EDTA) sodium salt via a thermal polymerization method [84]. g-CNQDs could effectively produce superoxide and hydroxyl radicals with the irradiation of visible light, eradicating ~99% *E. coli* and ~90% *S. aureus* at a concentration of 0.1 mg/mL. Moreover, g-CNQDs featured low cytotoxicity—3.2 mg/mL g-CNQDs were nontoxic to fibroblast cells. Liu et al. synthesized R-CDs through solvothermal treatment of 2,4-dihydroxybenzoic acid (an organic bactericide) and 6-bromo-2-naphthol [77]. R-CDs possessed both intrinsic antibacterial activity and antibacterial photodynamic activity toward multidrug-resistant *Acinetobacter baumannii* (MRAB). R-CDs could effectively enter bacterial cells and bacterial biofilms with few side effects on animal cells. Therefore, R-CDs were successfully utilized for MRAB biofilm prevention and elimination as well as the treatment of MRAB-induced infected wounds. No microbial drug resistance was observed when using R-CDs to kill MRAB and MRSA. These findings demonstrated that R-CDs represent a potent antibacterial agent for fighting against drug-resistant bacteria. Liu et al. fabricated zinc-doped CDs (Zn-CDs) from citric acid, ethylenediamine, and zinc acetate by a one-step hydrothermal method [87]. Zn-CDs produced ROS under blue light irradiation, showing bactericidal effect toward *S. aureus* and *Streptococcus mutans* with negligible animal cell toxicity. Collectively, the different types of CDs mentioned above can act as new types of photosensitizers for photodynamic antibacterial treatment.

Besides being used as photosensitizers, CDs can be integrated with other photosensitizers such as curcumin [90], black phosphorus (BP) nanosheets [91], TiO_2_ [92,93], and ZnO [68,94,95] to afford photoresponsive nanocomposites. In these photosensitive nanocomposites, CDs play different rols in obtaining improved antimicrobial PDT efficacy, such as being used as drug carriers [90], enhancing the interaction of photosensitizers with microorganisms [91,92], preventing the agglomeration of photosensitizers [92], or increasing the light absorption and suppressing photogenerated electron–hole’s recombination [95]. For instance, Yan et al. developed a nano-PS system using CDs to deliver the traditional photosensitizer curcumin (Cur) for enhanced antibacterial performance [90]. Zhang et al. decorated BP nanosheets with cationic CDs through in situ growth of CDs from chlorhexidine gluconate on the surface of BPs, resulting in the formation of the nanocomposite BPs@CDs (Figure 5A) [91]. Without light irradiation, BPs@CDs exhibited antibacterial ability due to the electrostatic attraction between the bacteria and the CDs on the surface of BPs@CDs. Under 660 nm laser irradiation, BPs@CDs produced singlet oxygen, exhibiting outstanding photodynamic antibacterial capacity. Under 808 nm laser irradiation, BPs@CDs displayed photothermal antibacterial activity. Accordingly, the BPs@CDs exhibited synergistic intrinsic antibacterial activity, antibacterial PDT activity, and antibacterial PTT activity toward both *E. coli* and *S. aureus*. In addition, BPs@CDs were degradable with no noticeable cytotoxicity. Owing to their triple-mode antibacterial capability, BPs@CDs were successfully deployed for the treatment of bacteria-associated wounds with shortened wound healing time.

The CD-involved antimicrobial PDT systems have been developed in the forms of different nanocomposites. Hydrophobic carbon quantum dots (hCQDs) with photodynamic property made from the poloxamer Pluronic F-68 [83] were encapsulated in polymers such as polydimethylsiloxane (PDMS) [96,97], polycaprolactone [98], and polyurethane [99], and to construct light-triggered antibacterial nanocomposites in the form of slide [96], nanofiber [98], and film [99]. For example, hCQDs were embedded into the PDMS polymer matrix to generate hydrophobic CQDs/PDMS surface by a swelling-encapsulation-shrink method [96]. The nanocomposite surface exhibited bactericidal activities against *S. aureus*, *E. coli*, and *K. pneumoniae* by producing ROS upon the excitation at 470 nm. More importantly, the nanocomposite surface showed no toxicity towards NIH/3T3 cells.

#### 2.4.2. Photothermal Therapy (PTT)

PTT eliminates microorganisms by hyperthermia generated from photothermal agents when they absorb light. Only one kind of antimicrobial CDs with PTT capacity had been reported when we prepared the previous review. Since 2019, more CDs have been reported for antimicrobial PTT [76,100,101]. Belkahla et al. produced CDs from glucose as the precursor through the alkali-assisted ultrasonic irradiation approach [100]. The generated CDs possessed the heat-producing capability under illumination at 680 or 808 nm and were employed for photothermal treatment of *E. coli*. Yan et al. constructed a nanosystem termed CDs-Tb-TMPDPA, which consisted of TMPDPA (4-(2,4,6-trimethoxyphenyl)-pyridine-2,6-dicarboxylic acid, a two-photon ligand)-sensitized Tb^3+^ as a temperature-sensitive module and CDs (prepared by microwave heating of citric acid and formamide) as a photothermal antibacterial component [101]. The CDs were coordinated to Tb^3+^ that was further linked with TMPDPA through coordination interaction. The authors demonstrated that the nanosystem could be used for temperature detection based on the temperature-dependent fluorescence intensity (*I*) ratio (*I*(Tb^3+^)/*I*(CDs)) and the fluorescence lifetime of CDs-Tb-TMPDPA. Besides, the authors also realized *E. coli* growth inhibition by utilizing the photothermal conversion property of CDs in CDs-Tb-TMPDPA via two-photon excitation (660 nm). This work develops a multifunctional probe for dual-mode temperature detection and antibacterial PTT under two-photon excitation.

Moreover, CDs-involved PTT can be integrated with other antimicrobial strategies such as PDT [90,93] and chemodynamic therapy (CDT) [102] to achieve combined antimicrobial therapies. For instance, N, S-doped CDs with strong fluorescence were first prepared from citric acid and thiourea by a one-step hydrothermal route, and was combined with curcumin (Cur) to afford CDs/Cur (Figure 5B) [90]. In CDs/Cur, the ROS yield of Cur could be enhanced through fluorescence resonance energy transfer (FRET), while the high photothermal conversion efficiency of the CDs due to their strong light absorption was preserved. As a result, upon 405 nm visible light and near-infrared light irradiation, CDs/Cur could yield ROS and a moderate temperature increase, which seriously damaged bacterial cell surface, leading to synergistic PDT- and PTT-promoted antibacterial effects against *E. coli* and *S. aureus*. Furthermore, CDs/Cur displayed low cytotoxicity and negligible hemolytic activity, which ensured their practical application. In another example, Yan et al. developed a nanocomposite termed FeOCl@PEG@CDs by coating poly(ethylene glycol) (PEG) and CDs on iron oxychloride nanosheets (FeOCl NSs) [102]. The hydroxyl radical (•OH) was generated from H_2_O_2_ activation by the redox cycle of ions on FeOCl NSs, and the heat was produced from the CDs upon the irradiation at 808 nm, inducing the death of *S. aureus* and *E. coli*. Further, the FeOCl@PEG@CDs were successfully applied in synergistic chemodynamic and photothermal treatment of infected wounds.

As far as we know, because only several CDs have been reported for photo-assisted antimicrobial uses, the lethal route was not carefully investigated. Commonly, researchers just reported that bacteria treated with CDs can produce ROS or heat upon light irradiation which can cause cell wall/membrane damage to kill bacteria. Thus, more studies should be performed in the future to thoroughly investigate the underlying lethal mechanism of CD-based antibacterial phototherapy.

## 3. Applications of Antimicrobial CDs in Medical and Industry Fields

CDs-involved antimicrobial strategies have been deployed in both medical and industry fields. In the medical field, antimicrobial CDs have been leveraged for coating the surface of orthopedic implant materials [66], delivering drugs [24,57,103,104], and repairing infected bone defects [47]. Moradlou et al. grew a thin film of CDs-incorporated hematite (CQDs@α-Fe_2_O_3_) on a titanium substrate to yield Ti/CQDs@α-Fe_2_O_3_ [66]_._ CQDs were prepared from graphite rods via an electrochemical method and used as nano-scaffolds for the growth of CQD@α-Fe_2_O_3_ nanoparticles as core@shell nanostructures. The Ti/CQDs@α-Fe_2_O_3_ samples exhibited sustainable antibacterial activity against *S. aureus* but not *E. coli*, offering a way of using CDs to prepare antimicrobial materials for medical devices. In another work, Geng et al. synthesized positively-charged CQDs (p-CQDs) through microwave reaction of spermidine trihydrochloride, and prepared negatively-charged CQDs (n-CQDs) via microwave reaction of 1,3,6-trinitropyrene (TNP) and sodium sulfite (Figure 6A) [47]. The p-CQDs displayed effective antibacterial activity against multidrug-resistant (MDR) bacteria and could realize the inhibition of biofilm formation, while n-CQDs notably promoted bone regeneration. The nearly neutral p-CQD/WS_2_ hybrids were first fabricated by depositing p-CQDs on WS_2_ nanosheets, and then coencapsulated with n-CQDs into the gelatin/methacrylate anhydride (GelMA) hydrogel to obtain p-CQD/WS_2_/n-CQD/GelMA hydrogel scaffold (Figure 6A). The implantation of p-CQD/WS_2_/n-CQD/GelMA hydrogel scaffold in an MRSA-infected craniotomy defect model induced almost complete repair of an infected bone defect with the new bone area of 97.0 ± 1.6% at 60 days. This work proposes a CD-based strategy for developing biomaterials with both antibacterial and osteogenic activities for the treatment of infected bone defects.

In the industry field, antimicrobial CDs have been utilized to construct thin-film composite membranes for forward osmosis [105] and nanofiller [106], packaging materials [107,108], and lubricant additives [53]. Mahat et al. developed thin-film composite membranes for forward osmosis by embedding CQDs derived from oil palm biomass into polysulfone-selective layers, which were denoted as CQDs-PSF (PSF: polysulfone) [105]. The authors proved that the addition of CQDs into PSF membranes increased water flux and improved antibacterial performance. In another study, Koulivand et al. constructed antifouling and antibacterial nanofiltration membranes for efficient salt and dye rejection by incorporating nitrogen-doped CDs (NCDs) to polyethersulfone (PES) using a phase inversion technique [106]. The antibacterial NCDs were synthesized via hydrothermal treatment of ammonium citrate dibasic. The obtained membrane exhibited improved pure water flux and enhanced antifouling property. In addition, Kousheh et al. constructed a nanocellulose film with antimicrobial/antioxidant and ultraviolet (UV) protective activities for food packaging by introducing water-dispersible and photoluminescent CDs [107]. The antimicrobial CDs were synthesized from cell-free supernatant of *Lactobacillus acidophilus* via a hydrothermal method. The as-synthesized CDs were embedded into bacterial nanocellulose (BNC) film due to the hydrogen bonding interaction between CDs and the carboxyl, hydroxyl, and carbonyl groups of BNC, leading to the formation of the CD-BNC film. The CD-BNC film displayed a higher inhibitory activity toward *Listeria monocytogenes* than *E. coli*. In addition to antibacterial activity, the introduction of CDs into the BNC film also endowed the CD-BNC film with UV-blocking activity, fluorescence appearance, and improved flexibility. The CD-BNC film could be used to fabricate nanopaper for wrapping of food commodity and fabrication of forgery-proof packaging. In addition to thin-film composite membranes and packaging materials, antimicrobial CDs have been implemented as lubricant additives. Tang et al. fabricated CDs from PEG and PEI through a hydrothermal approach [53]. The MICs of the CDs toward *E. coli* and *S. aureus* were 62.5 and 15.56 μg mL^−1^, respectively. Besides the antibacterial activity, the CDs featured anti-friction property. The addition of 0.2% (wt) CDs reduced the mean friction coefficient and wear volume of water-based lubrication by 59.77% and 57.97%, respectively. This example suggests that CDs with antibacterial and anti-friction functions can be utilized as an advanced lubricating additive, thus broadening the practical application of CDs.

## 4. Conclusions

As reviewed here, CDs are potent antimicrobial nanomaterials and represent promising alternatives to conventional antibiotics for the treatment of infectious diseases caused by microorganisms. Nevertheless, several challenges still exist. First, the potential antibacterial capability and specificity of CDs are difficult to predict from their raw materials, since it is unknown whether the antibacterial groups of raw materials can be retained or there are newly formed antibacterial structures during the complicated CD formation process. The CDs’ functional and biological characteristics are directly associated with their core and particularly their surface’s functional groups, which are largely dependent on the precursors and synthetic methods. Therefore, the structural analysis of CDs and the clarification of the reaction mechanism of CDs are helpful to better predict the antibacterial activities of CDs. Second, the integration of CDs with other compounds such as antibiotics, metal ions, hydrogels, and photosensitive materials to prepare new composite materials with synergistic antibacterial effect represents an important research direction in this field, which is definitely worth exploring in the future. Third, most antimicrobial CDs possess high MICs, and usually can only kill certain types of bacteria such as Gram-positive bacteria. Thus, it is highly desired to develop CDs with low MICs and broad-spectrum antibacterial ability. Fourth, the current antibacterial research of CDs mainly focuses on the killing of planktonic bacteria. In the future, it is necessary to explore the application potential of CDs in combating bacterial biofilms, eliminating intracellular bacteria, and killing bacteria in tumor. Fifth, despite the extensive investigations on the use of CDs for killing bacteria, few examples have been reported on using CDs to eliminate fungi and viruses which can also cause severe infections, diseases, and even death to humans. Sixth, the reproducible, large-scale, and cost-effective fabrication of CDs still limits the practical antimicrobial applications of CDs. Seventh, studies regarding the interaction of CDs with microbial cells, the distribution of CDs in microbial cells, the antimicrobial mechanisms of CDs, and the possible antimicrobial resistance development of CDs are still lacking, which will benefit the development of CDs with broad-spectrum antimicrobial activities, low MICs, and negligible drug resistance. Finally, although CDs are generally shown to be safe by cytotoxicity assays, the in vivo safety analyses of CDs remain largely unexplored, which is definitely worthy of evaluation in future studies. It is hoped that the current review will further promote the future design of functional CDs and CDs-incorporated advanced materials for combating the microbial infection-caused diseases.

## Data Availability

Data sharing not applicable.

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
