# Peer review of "Carbon Dots for Killing Microorganisms: An Update since 2019"

_pharmaceuticals, 2022, doi:10.3390/ph15101236_

Round 1

Reviewer 1 Report

The work is well written and complements the previous review on the use of carbon dots with antimicrobial properties written by the same team in 2019 and entitled “Carbon Dots for Sensing and Killing Microorganisms”- Journal of Carbon Research, 21 pages. The above publication was cited 56 times (2019-2, 2020- 19. 2021- 24 and 2022- 11, Web of Science). This indicates that the topic is important to the world of science, but not groundbreaking. In my opinion, the supplement (26 pages) submitted to me for review should also be well cited. I recommend this paper for publication.

Author Response

Response: We thank the reviewer’s time and effort in reviewing our manuscript. We want to thank the reviewer for stating that “The work is well written and complements the previous review…”, “the topic is important to the world of science”, and “In my opinion, the supplement (26 pages) submitted to me for review should also be well cited”. We also want to thank the reviewer’s recommendation of the publication of our paper.

Reviewer 2 Report

"Carbon dots for killing microorganisms: An update since 2019" by Fengming Lin, Zihao Wang and Fu-Gen Wu

This is a nice updated review. That said, it appears as a collection of references without much explanations. In my opinion, a good review ought to introduce the field for those who want to enter it; otherwise, a small Google search will accomplish the same goal.

1. The authors documented various fabrication methods and their applications, such as thermal and optically activated dots; yet, the question remains: why are CDs so lethal to pathogens while the micron-size particles are generally considered bio-compatible? A section devoted to the catalytic effects of small carbon, their crystallographic structure, the effect of surface states (either through defects, or by functionalization), charge transfer and oxidation mechanisms will enhance the write-up.

2. Lines 75-77: "For instance, Wang et al. synthesized CDs (ACDs) from Artemisia argyi leaves through a smoking simulation approach [34]. ACDs displayed selective antibacterial ability toward Gram-negative bacteria like Escherichia coli (E. coli),...."

- The author defined dots as smaller than 20 nm. I would add an SEM picture of ACDs to Fig. 1.

3. Fig. 1: "(C) Scanning electron microscopy (SEM) images of E. coli incubated without ACDs (a and b) and with ACDs (c and d), and S. aureus incubated without ACDs (e and f) and with ACDs (g and h)."

- What exactly do we see here? To the naked eye the dispersion of E. coli remained the same.  More description is needed.

4. Table 1: size of the various CD are quoted.  What exactly do the authors mean by size? If the dot is spherical, is it the radius of the dot? It is its narrow dimension (in case it was prepared from CNT)? Drawings, or even SEM pictures are recommended.  In case of non-spheroidal dots, which dimension is the active one and why?

5. Section 2.4 ought to be expanded to include the lethal route in photo and photo-thermic processes.  A drawing will also help

6. Finally, I would add a comparative section to illustrate the lethality of CD with respect to inorganic dots, such as TiO2, V2O5, Co3O4 or the like.

Author Response

Dear respected reviewer,

We have uploaded a separate file to reply to your very helpful and constructive suggestions.

Thank you so much!

Best wishes,

Fu-Gen Wu

Round 2

Reviewer 2 Report

The authors have answered all my queries. I recommend this paper for publication.

As a side note, the English should be re-checked and abbreviations should be stated in the beginning of the review (e.g., ROS= reactive oxygen species).